# The Hidden Impact of ADHD Symptoms in Preschool Children with Autism: Is There a Link to Somatic and Sleep Disorders?

**DOI:** 10.3390/bs14030211

**Published:** 2024-03-06

**Authors:** Dario Esposito, Katerina Bernardi, Arianna Belli, Valentina Gasparri, Sara Romano, Letizia Terenzi, Maria Elena Zanatta, Sara Iannotti, Mauro Ferrara

**Affiliations:** Department of Human Neurosciences, Sapienza University of Rome, 00185 Rome, Italy; dario.esposito@uniroma1.it (D.E.); katerina.bernardi@uniroma1.it (K.B.); arianna.belli@uniroma1.it (A.B.); valentina.gasparri@uniroma1.it (V.G.); sara.romano@uniroma1.it (S.R.); letizia.terenzi@uniroma1.it (L.T.); mariaelena.zanatta@uniroma1.it (M.E.Z.); sara.iannotti@uniroma1.it (S.I.)

**Keywords:** autism spectrum disorder, attention deficit hyperactivity disorder, somatic complaints, parental stress, sleep disorders, functional gastrointestinal disorders, preschoolers, preschool children

## Abstract

Neurodevelopmental disorders, such as autism spectrum disorder (ASD) and attention deficit hyperactivity disorder (ADHD), do often present in comorbidity among them and with other medical conditions, including sleep and gastrointestinal (GI) disorders and somatic complaints. An anonymous online survey based on standardized questionnaires (SDSC, KL-ASD, APSI, ROME V CRITERIA, CPRS, CBCL) was completed by the parents of 46 preschoolers diagnosed with ASD. A high prevalence (47%) of ADHD symptoms in this population was found, surpassing previous estimates. Sleep disturbances, especially difficulties in initiating and maintaining sleep and sleep–wake transition, are more pronounced in ASD patients with comorbid ADHD. Additionally, in patients over 4 years old, there is a high prevalence of functional GI symptoms (Rome V criteria: 38%). Parental stress is significantly associated with ADHD symptoms, sleep disorders, and GI symptoms. Comprehensive assessments considering both core symptoms of neurodevelopmental disorders and associated comorbidities are crucial for more effective treatment strategies and improved wellbeing in affected individuals and their families.

## 1. Introduction

Neurodevelopmental disorders, such as autism spectrum disorder (ASD) and attention deficit hyperactivity disorder (ADHD), are recognized to be associated with a broad spectrum of comorbidities, including sleep disorders and somatic disorders [1,2,3,4,5,6,7,8,9]. These conditions have a significant impact on both patients’ and parents’ quality of life.

### 1.1. Autism Spectrum Disorder: Definition, Incidence, and Comorbidities

ASD is a complex neurodevelopmental condition characterized by early-onset and persistent difficulties in social communication and interaction, as well as restricted interests and repetitive behavioral patterns [10]. The estimated worldwide prevalence of ASD among children is approximately 1 per 100 children, with a male-to-female ratio of 4.2:1 [11].

It is known that children with ASD have a higher prevalence of sleep disorders compared to typically developing children [2,3,4,12].

Looking at the recent literature, many studies revealed that several ASD comorbidities are significantly correlated with several sleep parameters, among which affective/anxiety problems showed the highest correlation with total sleep difficulties. Additionally, other studies have demonstrated that sleep disturbances in ASD are dimensionally associated with worse social impairment and poorer adaptive functioning [13].

The causal relationship between sleep disorders and ASD remains unclear. Various explanations have been proposed, including biological and genetic abnormalities [14], sleep problems as intrinsic aspects of ASD [15], and the possibility of them being a co-occurring condition independent from ASD [16]. Further research is necessary to understand the intricate relationship between sleep problems and psychiatric comorbidities in individuals with ASD [15]. 

In the same way, children with ASD, compared to those without a neurodevelopmental disorder, are up to five times more likely to develop feeding disorders (such as food selectivity, food refusal, and poor oral intake) and more than four-fold more likely to develop gastrointestinal (GI) symptoms, where constipation, diarrhea, and abdominal pain are reported most commonly [5,17].

GI disorders are also associated with increased ASD severity; for example, the odds of constipation increase with greater social impairment and poorer verbal abilities [18,19].

Moreover, GI conditions may cause discomfort and be an impediment to good sleep hygiene in children with ASD.

Although a reliable diagnosis of GI problems in ASD is important, recognition can be difficult due to presentations that lack classic signs of GI distress [5].

Maternal factors, genetics, diet, and specific microbiota environments are being investigated as potential factors contributing to disentangling the relationship between ASD behavior, cognition, and GI dysfunction [5,20].

### 1.2. Attention Deficit Hyperactivity Disorder: Definition, Incidence, and Comorbidities

On the other hand, ADHD is one of the most frequent neurodevelopmental disorders in children and adolescents, with a prevalence of 7.6% of 96,907 children aged 3 to 12 and 5.6% of teenagers [21,22]. According to DSM-5-TR, ADHD is characterized by a persistent pattern of inattention and/or hyperactivity–impulsivity that interferes with functioning or development [23]. Several problems commonly co-occur with ADHD, such as delays in language, motor, or social development, emotional dysregulation, or impulsivity. ASD is one of the most frequent neurodevelopmental comorbidities of ADHD [24].

Moreover, up to 70% of children with ADHD experience sleep problems, compared to 20–30% of healthy peers, with a greater prevalence of females [25].

The most frequent sleep problems include refusal to go to sleep, difficulty falling or staying asleep, and fractured sleep [3,26,27,28].

One of the main consequences is daytime sleepiness, which affects children’s functioning and quality of life, as well as worsening behavioral problems.

The causes of sleep disorders in these patients appear to be multiple. A 2020 study [29] highlighted how several factors may contribute to the quality of sleep: unhealthy sleep practices, comorbid conditions, shared biological components, or stimulant treatment side effects [29,30].

Nevertheless, gastrointestinal involvement seems to be significantly relevant in these patients, who have a greater probability of suffering from constipation and fecal incontinence [6].

### 1.3. ADHD and ASD: The Impact on Quality of Life and Parental Stress

Both ADHD and ASD have a significant impact on patients’ quality of life and parental stress. In recent years, in the field of neurodevelopmental disorders, there has been a growing emphasis on researching the quality of life (QoL). This aims to enhance wellbeing and identify associated symptoms that may contribute to the clinical presentation, other than the core symptoms. 

The QoL in ADHD is not only influenced by the main symptoms themselves but also by many factors, such as family context, friend network, and socioeconomic and cultural factors [8,31].

Several studies have reported a low quality of life in individuals with ASD [32]. However, there appear to be multiple factors involved, which makes it difficult to provide a more precise characterization, thus requiring further in-depth studies [9].

Children with ASD and ADHD, as previously discussed, often have co-existing GI and sleep issues [7]. These comorbidities may also impair their quality of life. 

Several studies have reported that parental stress is increased in families of children with ADHD or ASD, compared to that of parents of neurotypical children [33,34].

It is therefore important to consider both parental stress and the child’s QoL [35].

It has been discussed that ASD and ADHD comorbidity may negatively impact not only the quality of life and parental stress but also may increase the risk of physical problems.

Prevalence rates of ADHD in the context of autism (~22%) and autism in the context of ADHD (~21%) are significantly higher than those reported for children in the general population [36,37].

This overlap represents a real clinical challenge and enhances the importance of investigating the co-occurrence of physical symptoms in children with ASD and ADHD, to establish more adequate treatments.

The aspects analyzed so far are well studied and known in both adults and school-age children, while they appear to be less investigated in preschool children. 

The primary aim of this study was to assess the presence of ADHD symptoms and somatic complaints (including sleep and GI disorders) in preschool children with ASD and the influence on their quality of life. Moreover, also considering parental characteristics, our secondary aim was to evaluate the impact of these associated symptoms on parental stress. 

## 2. Materials and Methods 

### 2.1. Participants

For this study, we enrolled inpatients referred to our third-level center for neurodevelopmental disorders from October 2021 to December 2023. The inclusion criteria were age between 2 and 5 years and 11 months and a diagnosis of autism spectrum disorder (ASD). The diagnosis of ASD was made using the Autism Diagnostic Interview—Revised (ADI-R) [38], conducted with patients’ caregivers by an experienced clinician, and the Autism Diagnostic Observation Schedule II (ADOS-II) administered to the patients by trained examiners [39]. The exclusion criteria were families who did not provide contact information and families with an inadequate understanding of the Italian language. Based on these criteria, we initially enrolled a total of 89 patients.

The parents of the selected patients were primarily contacted by telephone and the aim of the study was explained, reaching a preliminary agreement. A specific survey, developed for this study, was sent to the parents that agreed.

Participants completed the anonymous online survey after reading the consent form written at the beginning and explicitly giving their consent to participate. The survey was developed and conducted following the guidelines set by the Checklist for Reporting Results of Internet E-Surveys (CHERRIES) [40].

There was no monetary or credit compensation for participating in this study.

Parents or caregivers of all subjects gave their informed consent for inclusion before they participated in this study. This study was conducted in accordance with the Declaration of Helsinki, and the protocol was approved by the local Ethics Committee (Project identification code #5365).

### 2.2. Measures

A (specific/self-developed) survey was arranged for this study.

The first section of the survey was devoted to sleep disturbances, assessed using the Sleep Disturbance Scale for Children (SDSC) validated for preschool children. It consists of 26 items on a five-point Likert-type scale (1 = least severe and 5 = most severe). This 26-item, Likert-scale, parent-rated questionnaire measures sleep disturbance across six dimensions: sleep breathing disorders (SBDs); disorders of excessive somnolence (DOESs); difficulty in initiating and maintaining sleep (DIMS); sleep–wake transition disorders (SWTD); disorders of arousal (DoA); and sleep hyperhidrosis (SH). The sum of the single items’ scores provides a total sleep score ranging from 26 to 130. A total sleep score of >70 is considered pathological (internal consistency: 0.71–0.79; test–retest reliability: 0.71; diagnostic accuracy: 0.91; sensitivity: 0.89; specificity: 0.74) [41].

Health-related quality of life was investigated using the KidsLife scale for children with ASD and/or intellectual disabilities (KL-ASD). The main goal of the KL-ASD is to assess the quality-of-life-related personal outcomes in the childhood and adolescence of persons with significant disabilities. The field-test version of the scale is composed of 156 items on a 4-point frequency scale (0 = “never” and 4 = “always”). All items were divided into eight subscales: emotional wellbeing, material wellbeing, physical wellbeing, personal development, rights, self-determination, social inclusion, and interpersonal relationships. Higher scores indicate a better quality of life. For the purpose of this study, we exclusively used the physical wellbeing domain of this scale (internal consistency: 0.88) [42].

Caregivers also completed the Autism Parenting Stress Index (APSI), a measure of parental stress perceived by parents/guardians. It reflects the physical, social, and communication barriers imposed by the disability. As such, the 13 items on this test fall into three categories: core social disability, difficult-to-manage behavior, and physical issues. The answer format is a 5-point frequency scale from 0 = “not stressful” to 5 = “so stressful sometimes we feel we can’t cope” (internal consistency: 0.83) [43].

Gastrointestinal functional symptoms were investigated with the Rome criteria, which provide symptom-based guidelines by which child and adolescent functional gastrointestinal disorders (FGIDs), such as irritable bowel syndrome (IBS), functional dyspepsia, and rumination syndrome, can be diagnosed. The reliability of the presence of an FGID diagnosis was considered moderate, with an agreement with expert clinician judgment of 83.1% and a kappa value of 0.61 (sensitivity: 0.75; specificity: 0.90) [44].

For the purpose of this study, we also collected some questions from the Illness Attitude Scale (IAS) and selected and adapted them into a form for parents. The IAS has nine sub-scales as follows: worry about illness (W); concerns about pain (CP); health habits (HH); hypochondriacal beliefs (HB); thanatophobia (Th); disease phobia (DP); bodily preoccupations (BP); treatment experiences (TE); and effects of symptoms (ES). Each of the main IAS scales is scored on a five-point Likert scale from 0 to 4, with the respondents answering the questions as they pertain to themselves (test–retest reliability: 0.80; internal consistency: 0.85) [45].

Finally, the attention deficit hyperactivity disorder symptoms were estimated through the Conners Parental Comprehensive Behaviour Rating Scale (CPRS), a tool used to gain a better understanding of academic, behavioral, and social issues that are seen in young children. The questions are multiple-choice; examples of the topics of these questions relate to content scales: emotional distress, aggressive behaviors, academic difficulties, hyperactivity/impulsivity, separation fears, violence potential, and physical symptoms. For the purpose of this study, we used the short parental version of the CPRS, consisting of 25 questions with the possibility to vary depending on the child (internal consistency: 0.89; inter-rater agreement: 0.75) [46].

Behavioral problems were investigated through the Child Behavior Checklist for ages 1.5–5 years (internal consistency: 0.86; inter-rater reliability: 0.80; test–retest reliability: 0.85) [47].

### 2.3. Data Analysis

Data are expressed as mean ± standard deviation and range (minimum and maximum values) for continuous variables and as counts and percentages for categorical variables. Efforts were made to prevent missing data by thorough examination of clinical documentation. Additionally, an online user-friendly form was utilized, administered in a single session with mandatory responses, further reducing the likelihood of missing data. However, despite these measures, rare instances of missing data at random (MAR) persisted. To handle these remaining cases, a listwise deletion approach was employed, wherein cases with missing data were simply omitted from the analysis. For group comparisons, we employed chi-square tests or Fisher tests (in case of expected frequencies < 5) for categorical variables and T-tests for continuous data. For comparisons of continuous variables not normally distributed, Mann–Whitney U-tests were used instead of T-tests. The Shapiro–Wilk method was employed for normality tests. Given the dimensions of the sample and the type of distribution, Spearman’s correlation coefficient was used to measure the strength and direction of the relationship between variables. Linear regression models were created to estimate adjusted odds ratios (ORs; 95% confidence interval [CI]) of parental stress scores (dependent variable) in relation to other clinical characteristics (independent variables). For all comparisons, *p*-values less than 0.05 were considered to be statistically significant. Statistical analyses were performed using the Jamovi statistical software (version 2.3.28) based on the R language [48,49,50,51].

## 3. Results 

### 3.1. Description of the Sample

Out of the 89 inpatients selected to participate in our study, 9 patients were excluded as their parents did not provide consent to participate. The parents of 46 out of 80 remaining patients actively engaged in answering the questionnaires of our study. The final sample was composed of 46 patients (8 females and 38 males). Thirty were more than 4 years old (65.2%) and sixteen were less than 4 years old (34.7%). In Table 1, we report the description of the scores obtained in the most relevant subscales included in the questionnaire. 

Regarding age differences, patients aged more than four had statistically more oppositional behaviors (oppositional subscale of CPRS, *p* = 0.043) and more FGID diagnoses (*p* = 0.007). 

On the other hand, parents of patients aged less than 4 years worried more about their children’s health (worry about illness subscale of IAS, *p* = 0.050).

Concerning gender, we found no statistically significant differences except for higher levels of cognitive problems/inattention in females (*p* = 0.004).

### 3.2. Clinical Features Associated with ADHD Symptoms

In total, 54.3% of our patients obtained a clinical score for ADHD in at least one questionnaire among the CBCL (DSM-oriented ADHD subscale) and/or CPRS (ADHD index subscale). In particular, 13% of the patients obtained a clinical score in the CBCL, while 52.2% achieved a CPRS clinical score; only 10.9% of the patients had clinical scores in both questionnaires. 

Table 2 shows a comparison of the scores obtained in some relevant subscales of ADHD and non-ADHD patients (based on the CPRS). We found a significant difference in sleep disorders (*p* = 0.045), particularly in the SDSC subscales of initiating and maintaining sleep and sleep–wake transition disorders (respectively *p* = 0.023 and 0.022), parental stress level (*p* = 0.001), internalizing (*p* = 0.002), externalizing (*p* = 0.008), and behavioral problems (*p* < 0.001), as well as aggressive behavior (*p* = 0.015), affective problems (*p* = 0.006), withdrawal (*p* < 0.001), and somatic complaints (*p* = 0.012).

### 3.3. Physical Complaints and Functional Gastrointestinal Disorders

With regard to the patient’s general health, we analyzed somatic symptoms and the possible presence of a functional gastrointestinal disorder. In particular, 26.1% (12 out of 46) were diagnosed with a functional gastrointestinal disorder (FGID) according to the Rome V criteria, with one of them receiving two FGID diagnoses at the same time. Moreover, with regard to the CBCL somatic complaints subscale, 5.1% of the patients achieved borderline scores and 2.6% clinical scores (Table 3). 

### 3.4. Clinical Features Associated with Parental Stress 

With regard to parental stress, a significant association was found between APSI scores and the following variables through a multivariate linear regression model (Table 4): oppositional index and ADHD index of CPRS, somatic complaints index of CBCL, worry about illness index of IAS, disorder of initiating and maintaining sleep index, and sleep–wake transition disorder index of SDSC. In particular, we found that this model explains 68.5% of the variance in APSI scores (R^2^ = 0.759, adjusted R^2^ = 0.685).

As shown in Table 5, the parental stress index (APSI score) showed a significant positive correlation with the level of concern about illnesses (IAS) and the ADHD index. Both the APSI score and ADHD index showed a significant positive correlation with sleep disturbances, somatic complaints, and the withdrawal index of the CBCL. We also found that the presence of somatic complaints had a significant positive correlation with the withdrawal index and functional gastrointestinal disorders. Other significant correlations are reported in Table 5.

## 4. Discussion

The present study delves into the impact of ADHD symptoms in preschool children with ASD on some important comorbidities, namely sleep disorders, gastrointestinal (GI) issues, somatic complaints, and parental stress.

A recent study [52] found ADHD symptoms in 22% of preschool children using the DSM-oriented ADHD subscale of a broader questionnaire (CBCL) (cut-off T-score > 65). 

However, using the CBCL, this prevalence is lowered to 13% of children with clinical ratings (T-scores > 70) or 19.5% considering all elevated T-scores (>65). 

In our sample, we found a prevalence of more than 47% of ADHD symptoms using specific questionnaires for attention, hyperactivity, and behavioral problems (CPRS). 

The difference between the prevalence according to the two different tests might be attributed to the fact that the Conners rating scales (CPRS) are more sensitive to behavioral problems than a broader questionnaire such as the CBCL [53]. Moreover, it is worth noting that we considered ADHD symptoms as reported by caregivers, while the diagnosis of ADHD requires the presence of these symptoms in more contexts [10].

In contrast with attention deficit and hyperactivity symptoms, oppositional behaviors showed a significant difference between patients aged more than 4 years and younger children (*p =* 0.043). In younger children, oppositional behaviors might be underestimated by caregivers, being more often judged developmentally “normal” and more easily managed by adults, leading to less distress [54,55].

On the other side of the spectrum of behavioral difficulties, previous studies have shown that patients with ASD have higher scores on the CBCL withdrawal subscale, which might be associated with the core autistic symptom of social communication impairment [56]. Similarly, in the population of patients with ADHD, high withdrawal scores have been detected, especially in association with oppositional behaviors [57]. From our comparison between ASD children with and without ADHD symptoms, parent-reported withdrawal scores are more common in those with elevated ADHD symptoms (*p* < 0.001). This could be due to parents perceiving their child as more withdrawn because of greater difficulties in social interactions caused by reduced attention spans and motor instability. Another possible explanation is that deficits in joint attention may mediate the association between high scores in withdrawal scales and ADHD scales [58].

Regarding sleep, it is known that both ASD and ADHD exhibit notably elevated rates of comorbid sleep disturbances [37]; 17.4% of our preschool children sample had elevated scores of sleep difficulties according to a standardized questionnaire (SDSC), which is coherent with a recent Italian survey on a comparable population (18%) [59].

However, through subgroup comparisons, we found that autistic children with elevated ADHD symptoms showed a higher prevalence of sleep disorders compared to those with ASD alone (*p =* 0.045). It should be noted that a recent study implies no discernible difference in the prevalence of sleep disorders when comparing school-aged children with both ASD and ADHD with those with ADHD alone [60]. This discrepancy may suggest that ADHD might be a stronger determinant of sleep disturbances than ASD. 

Moreover, in our study, not all sleep features were significantly altered in these patients. In particular, the SDSC subscales of sleep–wake transition disorders were higher in children with comorbid ADHD; this fits with data from the literature, since it is known that ADHD has an increased correlation with sleep movement disorders such as restless leg syndrome, periodic limb movement disorders [61], or bruxism [62].

Also, ADHD may exert a cumulative effect on initiating and maintaining sleep difficulties, given that these sleep disturbances are also predominant in patients with ASD alone [63]. This may highlight the importance of overcoming the consideration of child sleep problems as a unitary construct. However, it is worth contemplating that poor sleep significantly impacts daytime behavior in terms of irritability, anxiety, and hyperactivity [64].

Despite a growing body of studies that acknowledge and describe the sleep issues associated with ADHD, it is complex to define the possible basic neurobiological mechanisms that are in common: the correlation appears to be intricate, involving a multiform interaction among various neurotransmitter systems (e.g., cholinergic system, monoaminergic pathways), nuclei and neurocircuitry in a system that controls both sleep and wakefulness (e.g., Ventrolateral preoptic nucleus), and delayed onset of melatonin secretion at night [30].

Furthermore, our study found a direct correlation between sleep disorders in ASD patients and elevated parental stress, aligning with prior reports indicating that a child’s sleep problems detrimentally affect parental sleep and impose considerable stress on family life [65].

Concerning GI disorders in children with ASD, the available literature presents conflicting results, with a prevalence ranging from 9% to 70% or higher [17].

A Norwegian study analyzing 195 toddlers found that GI symptoms were more common in children with ASD compared to those with typical development or other developmental delays. In particular, they found a prevalence of GI symptoms of around 53% in children between 18 and 36 months [66]. In a USA sample, parents reported significantly more GI problems in children with ASD compared to their unaffected siblings (42% vs. 12%) [67]. Another Italian study conducted on a sample of 115 preschoolers with ASD reported that 37% of them suffered from GI symptoms [68]. Another study found that 200 ASD children aged 2–5 years presented constipation in 64%, diarrhea in 32%, nausea/vomiting in 21%, and stomachache/pain in 45% [19].

Nevertheless, when analyzing GI disorders according to specific diagnostic criteria, the prevalence might change, while being more accurate across different studies. The most used diagnostic criteria for non-structural GI conditions (also known as functional gastrointestinal disorders (FGIDs) or—more recently—disorders of gut–brain interaction) are the Rome Foundation criteria [69].

Based on the Rome criteria, approximately 22–25% of normally developing children fulfill symptom-based criteria for an FGID [70].

On the other hand, a recent meta-analysis estimated a higher prevalence (33%; 95% CI, 13%–57%) of FGIDs in children aged 4–18 with ASD according to the Rome criteria [71].

In our sample, we observed a total prevalence of 26% of patients with ASD exhibiting FGIDs, which is consistent with those found in the general pediatric population. However, when focusing only on children more than 4 years of age, the diagnoses of FGIDs became more frequent, up to 37.9%, which is consistent with the higher prevalence of these diagnoses in older children with ASD [71].

This association between behavioral problems in ADHD and GI disorders could have several potential explanations, such as altered communication between the central nervous system and the enteric nervous system, responsible for an alteration in GI mobility and the perception of distension; poor responses to the stimulus of defecation due to behavioral problems; and nutritional deficiency [6]. Further research is needed to understand why the prevalence of FGIDs increases only in older children with ASD. Several points should be considered, in particular the different manifestations of GI symptoms across ages, the adequacy of diagnostic tools and questionnaires in younger age groups, the role of structural comorbid disorders in children with ASD, and the possible influence of parental stress and anxiety on somatic symptoms, including FGIDs.

Most of the above-mentioned symptoms and comorbidities have been thoroughly investigated in the literature due to their relevant impact on distress and the quality of life of parents of children with ASD [32,35,72,73]. Notwithstanding, few studies have explored the combined effects of different conditions and physical symptoms on parental stress in this kind of population and age group. Interestingly, linear regression models showed that high levels of parental stress were correlated with ADHD symptoms, oppositional behaviors, somatic complaints, sleep disturbances, and parental concern about illness. It is interesting that the key features that influence parental stress levels are not the core symptoms of ASD but rather the associated symptomatology and resulting comorbidities [74]. It has been established, indeed, that co-occurring conduct problems such as oppositional, aggressive, and externalizing behavior are important predictors of parenting stress [72].

The parents’ emotional burden arising from this symptomatology is also likely related to parental intrinsic characteristics. Parental disturbances or sub-clinical traits (such as ASD and ADHD symptoms) may increase parenting stress. In addition, parents of children with ASD and ADHD show more depressive symptoms compared with normal data [75].

Furthermore, the parental tendency to experience health anxiety “by proxy” (worrying for their children’s health) is another crucial factor contributing to their stress [33]. Our results confirm this role and also show that worries about children’s health were more frequent in parents of younger children (<4 years of age). This may be explained by the less developed communication skills of the child or by a shorter parental experience in recognizing the child’s health problems and estimating their severity.

### Study Limitations and Future Research Prospects

This study has some limitations that we would like to disclose. Firstly, we used parent report measures to evaluate the symptoms of interest, and—despite the questionnaires of our choice being well validated for the selected sample—this could have led to report bias. Future studies may therefore consider implementing direct observations or instrumental assessment when evaluating ADHD symptoms, sleep disturbances, and somatic symptoms in children with ASD. In addition to that, our sample was gender-imbalanced, with a small number of females, and this may have affected findings concerning differences between males and females. However, this is a common limitation of research regarding ASD: even studies that better account for ascertainment bias and issues of diagnostic instruments still show a 2–5:1 male predominance in prevalence of ASD [76]. Given this imbalance, multicentric studies may be considered to obtain larger female samples. Furthermore, with regard to the age differences of children in our preschool sample (with most children being more than 4 years old), it is worth noting that this discrepancy has also been discussed elsewhere in the literature. A greater number of (stable) ASD diagnoses are indeed reported in older preschool children, which might have contributed to this discrepancy in our study as well [77].

Finally, the sample included in our research was not population-based but consisted of children brought to clinical attention. This may possibly have led to a selection bias, with a greater rate of severely affected children and a smaller number of individuals with milder symptoms included in this study. Future studies may consider selecting participants from the general population to obtain findings that can be more reliably generalized to the entire clinical population of choice.

This study has useful implications for clinical practice. A better awareness of the comorbidities presented by children with ASD is crucial to encourage clinicians to screen patients for them during clinical assessment. Moreover, the findings regarding the factors influencing parental stress are paramount to improve caregivers’ quality of life and prevent secondary negative outcomes, such as parental depression and anxiety disorders. This may, in turn, positively affect parent–child interactions and parental involvement in rehabilitation programs, factors that play a pivotal role in improving their children’s outcomes.

## 5. Conclusions

In conclusion, this study provides insights into the comorbidities in children with ASD and the impact they have on parental stress. Our findings show that ADHD symptoms are very common in preschool-aged children with ASD, with a prevalence ranging from 13 to 47%. Sleep disturbances and gastrointestinal issues are also prevalent, especially in children with comorbid ADHD symptoms. This study also highlights the impact of comorbidities on parental stress, emphasizing the importance of assessing and addressing parental wellbeing in the management of neurodevelopmental disorders. Overall, a comprehensive assessment that considers core symptoms, associated comorbidities, and parental characteristics is crucial for effective treatment strategies and improved wellbeing in affected individuals and their families.

## Figures and Tables

**Table 1 behavsci-14-00211-t001:** Mean and standard deviation, divided by gender (F: female, M: male), of age in months and the following subscales: (physical wellbeing—KidsLife ASD (KL-ASD), Autism Parenting Stress Index (APSI), Sleep disorders—Disturbance Scale for Children (SDSC), Parental Health Anxiety by proxy (IAS); Parental fear of pain (IAS); treatment experiences (IAS), ADHD index of the Conners Comprehensive Behaviour Rating Scale (ADHD ^a^), Total problems scale (CBCL), Withdrawal scale (CBCL).

	Gender	Mean	SD
Age (months)	F	54.49	10.792
	M	51.66	12.171
Physical wellbeing (KL-ASD)	F	24.28	6.993
	M	21.15	6.732
Parental stress (APSI)	F	12.42	7.976
	M	13.66	8.112
Sleep disorders (SDSC)	F	52.29	7.544
	M	55.55	11.420
Parental health anxiety by proxy (IAS)	F	3.71	4.536
	M	6.24	3.859
Parental fear of pain (IAS)	F	4.71	1.113
	M	6.29	2.847
Treatment experiences (IAS)	F	5.00	1.155
	M	4.89	1.721
ADHD ^a^	F	82.14	24.155
	M	64.24	12.467
Total problems (CBCL)	F	54.67	16.428
	M	57.94	10.241
Withdrawal (CBCL)	F	70.0	12.681
	M	67.87	9.101

**Table 2 behavsci-14-00211-t002:** Comparison between ADHD patients and non-ADHD patients. We report the mean and standard deviation of the scores and the *p*-value obtained using an independent sample T-test. * statistically significant. ^a^ According to CPRS “ADHD Index” subscale scores. Abbreviations: APSI, Autism Parenting Stress Index; CBCL, Child Behavior Checklist; CPRS, Conners Parental Rating Scales; KL-ASD, KidsLife quality of life scale—Autism Spectrum Disorder; IAS, modified Illness Attitude Scale; SDSC, Sleep Disturbance Scale for Children.

	Non-ADHD ^a^		ADHD ^a^		
	Mean	DS	Mean	DS	*p*
Age (months)	52.8	11.6	51.4	12.4	0.704
Physical wellbeing (KL-ASD)	19.9	7.1	23.3	6.2	0.103
Parental stress (APSI)	9.7	7.1	17.1	7.2	0.001 *
Sleep Disorders (SDSC)	51.7	75	58.2	12.7	0.045 *
Parental health anxiety by proxy (IAS)	5.1	3.8	6.5	4.2	0.253
Parental fear of pain (IAS)	6.2	2.6	5.9	2.8	0.639
Treatment experiences (IAS)	4.4	1.4	5.3	1.7	0.066
Internalizing problems (CBCL)	52.7	9.6	61.8	7.9	0.002 *
Externalizing problems (CBCL)	48.7	10.2	60.2	10.8	0.008 *
Total problems (CBCL)	50.6	9.3	63.3	9.2	<0.001 *
Aggressive behavior (CBCL)	52.8	5.0	59.9	10.7	0.015 *
Emotionally reactive (CBCL)	54.2	5.0	58.1	8.0	0.076
Affective problems (CBCL)	53.7	5.2	59.5	6.9	0.006 *
Anxiety/Depression (CBCL)	52.7	3.11	56.6	7.0	0.033
Somatic complaints (CBCL)	51.3	2.2	56.0	6.6	0.012 *
Withdrawal (CBCL)	62.8	8.8	73.1	7.4	<0.001 *

**Table 3 behavsci-14-00211-t003:** Percentage of patients divided between ages above and below 4 years old who received a diagnosis of a functional gastrointestinal disorder (using Rome V criteria) or obtained clinical scores for sleep disorders (using the Sleep Disturbance Scale for Children (SDSC) and somatic complaints (using CBCL). ^1^ Somatic complaints index of CBCL. * % of patients (out of 46). ** % calculated on the age subgroups.

Diagnosis	Age (Years)	% per Age **	% of Total *
Functional constipation	<4	7.1%	13.0%
	>4	16.1%	
Functional dyspepsia	<4	0	2.2%
	>4	3.2%	
Fecal incontinence	<4	0	13.0%
	>4	19.4%	
Sleep disorders	<4	13%	17.4%
	>4	4.3%	
Somatic Complaints ^1^	<4	0	2.6%
	>4	2.6%	

**Table 4 behavsci-14-00211-t004:** Clinical variables associated with Autism Parenting Stress Index (APSI) (*N* = 46): linear regression model.

Independent Variables	Linear Regression Model		
	*p*	SE	95% C.I.
Age (months)	0.221	0.126	−0.08–0.33
Sex, Male vs. Female ^a^	0.575	0.186	−0.48–0.85
Oppositional index of CPRS	**<0.001**	0.524	0.23–0.81
ADHD index of CPRS	**0.029**	0.410	0.04–0.77
Hyperactive–Impulsive index of CPRS	0.125	−0.273	−0.62–0.08
Somatic complaints index of CBCL	**0.004**	0.322	0.11–0.53
Worry about illness index of IAS	**<0.001**	0.481	0.26–0.70
Disorder of initiating and maintaining sleep index of SDSC	**0.010**	0.396	0.10–0.69
Sleep–wake transition disorder index of SDSC	**<0.001**	0.609	0.28–0.93

^a^: reference level. Bold indicates significant values (*p* < 0.05). Multivariate model fit measures: (R^2^ = 0.759, adjusted R^2^ = 0.685), VIF < 4.0 for all the independent variables.

**Table 5 behavsci-14-00211-t005:** Correlation matrix. ^a^ According to CPRS “ADHD Index” subscale scores. Abbreviations: APSI, Autism Parenting Stress Index; CBCL, Child Behavior Checklist; CPRS, Conners Parental Rating Scales; KL-ASD, KidsLife quality of life scale—Autism Spectrum Disorder; IAS, modified Illness Attitude Scale; SDSC, Sleep Disturbance Scale for Children.

	Age (Months)	Parental Stress (APSI)	ADHD ^a^	Somatic Complaints (CBCL)	Withdrawal (CBCL)	Sleep Disorders (SDSC)	Parental Health Anxiety by Proxy (IAS)	Treatment Experiences (IAS)	Functional Gastrointestinal Disorder
Age (months)	—								
Parental stress (APSI)	−0.064	—							
ADHD ^a^	−0.138	0.517 ***	—						
Somatic complaints (CBCL)	−0.068	0.390 *	0.327 *	—					
Withdrawal (CBCL)	−0.079	0.372 *	0.600 ***	0.432 **	—				
Sleep Disorders (SDSC)	−0.062	0.372 *	0.403 **	0.236	0.464 **	—			
Parental health anxiety by proxy (IAS)	−0.103	0.485 ***	0.229	0.296	0.076	0.290	—		
Treatment experiences (IAS)	−0.005	0.219	0.296 *	0.031	0.013	0.123	0.258	—	
Functional gastrointestinal disorder	0.330 *	0.104	0.153	0.348 *	0.187	0.066	−0.034	0.041	—

Note. * *p* < 0.05, ** *p* < 0.01, *** *p* < 0.001.

## Data Availability

The data that support the findings of this study are available on request from the corresponding author.

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
