# Peer review of "The Hidden Impact of ADHD Symptoms in Preschool Children with Autism: Is There a Link to Somatic and Sleep Disorders?"

_behavsci, 2024, doi:10.3390/bs14030211_

Round 1

Reviewer 1 Report

Comments and Suggestions for Authors

Thank you for  the opportunity to review this important and timely manuscript. ASD and ADHD are rising in prevalence and research devoted to understanding the syndrome of these conditions along with common symptoms that hinder QOL (Sleep quality, GI issues) is highly valuable. The authors did a commendable job of studying ASD, ADHD, and ASD + ADHD conditions and presenting the results in an effective manner. I believe this manuscript would be a good contribution to this special issue.

Comments on the Quality of English Language

NA

Author Response

We thank the Reviewer for taking the time to analyze our manuscript. We greatly appreciate the positive feedback and recognition of the importance of our study on ASD, ADHD, and their impact on quality of life, particularly regarding sleep quality and GI issues.

Reviewer 2 Report

Comments and Suggestions for Authors

Author Response

Please see below, in italics, for a point-by-point response to the reviewers’ comments and concerns.

This study investigates the impact of Attention-Deficit/Hyperactivity Disorder (ADHD) symptoms on comorbidities in preschool-aged children diagnosed with autism spectrum disorder (ASD). The study examines the prevalence of ADHD symptoms and their association with sleep disturbances, gastrointestinal (GI) issues, somatic complaints, and parental stress. Findings reveal a high prevalence of ADHD symptoms in preschool children with ASD, ranging from 13% to 47%. Children with comorbid ADHD symptoms exhibit a higher prevalence of sleep disorders and GI issues. The study highlights the significant burden of comorbidities on parental stress and underscores the importance of comprehensive assessment and management strategies for improving the well-being of children with ASD and their families. 

The introduction covers a wide range of topics related to neurodevelopmental disorders, focusing on autism spectrum disorder (ASD) and attention deficit hyperactivity disorder (ADHD). While the information provided is relevant, the introduction lacks a clear structure. Consider breaking down the content into subsections to improve readability. Ensure smooth transitions between different topics and paragraphs to maintain coherence and logical flow (the transition between discussing ASD and ADHD, for example). Ensure that each point directly contributes to framing the study's objectives and research questions. Some sections, such as the discussion on parental stress and quality of life, seem tangential to the primary focus of the study on preschool patients with ASD. Highlight why investigating ADHD symptoms, somatic symptoms, and parental stress in preschool patients with ASD is important and how it contributes to existing knowledge gaps.

  • Thank you for your feedback regarding the introduction's structure and coherence. We have revised the introduction to include clear subsections that better delineate the topics discussed. We have also acknowledged your point about clarifying the relevance and importance of investigating ADHD symptoms, somatic symptoms, and parental stress in preschool patients with ASD. 

Provide more details on how the 89 inpatients aged 2 to 5 years and 11 months were initially enrolled. Explain any inclusion or exclusion criteria used to ensure the sample's representativeness and relevance to the study's objectives. While it's mentioned that ethics approval was not required according to Italian Research Ethics guidelines, it's essential to provide a brief explanation of why this exemption was granted. Clearly state how the study ensured participant confidentiality, anonymity, and voluntary participation to adhere to ethical standards. 

  • We have provided more detailed information on participant enrollment, including any inclusion or exclusion criteria used to ensure sample representativeness and relevance to the study's objectives. Additionally, we have modified the section regarding the Institutional Review Board that erroneously reported that ethics approval was not required (as previously discussed with the journal editor). 

For each measurement tool used in the study (e.g., Sleep Disturbance Scale for Children, KidsLife scale, Autism Parenting Stress Index), briefly explain its reliability, validity, and relevance to the study's objectives. Mention how missing data, if any, were handled during the data analysis process.

  • In the revised manuscript, we have included brief explanations of the main psychometric properties of each measurement tool used in the study. We have also included a section dedicated to missing data prevention and management.

The discussion effectively compares the prevalence rates of ADHD symptoms in the current

study with those reported in previous literature. It would be beneficial to discuss any potential

reasons for the variation in prevalence rates observed across studies. The discussion on sleep disturbances and GI disorders provides insightful interpretations of the study findings, but it would be beneficial to discuss potential mechanisms underlying the association between ADHD symptoms and these comorbidities, such as shared underlying neurobiological mechanisms or common risk factors.

  • We appreciate your feedback on comparing prevalence rates of ADHD and providing insightful interpretations of the study findings. In our revision, we have expanded the discussion about potential reasons for variation in ADHD prevalence rates observed across studies, mainly due to diagnostic tools differences and population characteristics. Moreover, we have discussed potential neurobiological mechanisms underlying the association between ADHD symptoms and comorbidities like sleep disturbances and GI disorders, including common risk factors.

The conclusions effectively summarize the key findings of the study and emphasize the importance of comprehensive assessment and management strategies for children with ASD and comorbid ADHD symptoms. Briefly discuss the implications of the study findings for clinical practice and future research directions.

  • We have expanded on the implications of our study findings for clinical practice and future research directions in an appropriate new paragraph at the end of the discussion section. 

We thank the reviewer for the constructive feedback, which we have incorporated into our revision to enhance the overall quality of the study.

Reviewer 3 Report

Comments and Suggestions for Authors

The manuscript addresses, through a comparative cross-sectional design, the association of ADHD symptoms with sleep disorders and somatic symptoms, especially gastrointestinal, in preschoolers with autism. The high ADHD-ASD comorbidity and the insufficient studies in preschool children justify the need to study this topic.

This is a necessary clinical study, whose strong point is the validity of the diagnoses involved, based on clinical interviews with the application of diagnostic criteria (DSM-5-TR, Rome V).

I want to make some comments to the authors.

My main concern regarding the research is the representativeness of the sample, and not so much because of its size, but because of the discrepancy between the number of boys (38) and girls (8), and their ages (30 over 4 years compared to 16 children under 4 years old), taking into account that analyses based on sex and age were carried out, and that conclusions are also established on the total sample. How do the authors think these circumstances may have affected the results? The difficulty of finding clinical samples is considered, but at least the authors should justify these deficiencies or declare them as limitations.

My main concerns regarding the manuscript are:

- No limitations or future lines of research are included in the discussion.

- The tests' reliabilities are not reported in the Measures section.

Other minor concerns are:

- In the keywords, I would replace physical health with somatic complaints, and preschool, toddler, and children with preschoolers or preschool children.

- In Measures, the reference Schalock and Verdugo (2002) does not comply with the citation style and does not indicate the 4-point range of scores (1-4?, 0-3?).

- Regarding the variables, I recommend homogenizing the use of gender or sex, but not using them interchangeably.

- In Results, I miss a table with the results of the MLR that allows us to assess the weight of the predictor variables.

- In lines 241-242 it would be good to comment on the most significant correlations.

Author Response

Please see below, in italics, for a point-by-point response to the reviewers’ comments and concerns.

The manuscript addresses, through a comparative cross-sectional design, the association of ADHD symptoms with sleep disorders and somatic symptoms, especially gastrointestinal, in preschoolers with autism. The high ADHD-ASD comorbidity and the insufficient studies in preschool children justify the need to study this topic.

This is a necessary clinical study, whose strong point is the validity of the diagnoses involved, based on clinical interviews with the application of diagnostic criteria (DSM-5-TR, Rome V).

  • Thank you for your feedback

I want to make some comments to the authors.

My main concern regarding the research is the representativeness of the sample, and not so much because of its size, but because of the discrepancy between the number of boys (38) and girls (8), and their ages (30 over 4 years compared to 16 children under 4 years old), taking into account that analyses based on sex and age were carried out, and that conclusions are also established on the total sample. How do the authors think these circumstances may have affected the results? The difficulty of finding clinical samples is considered, but at least the authors should justify these deficiencies or declare them as limitations.

  • We acknowledge your concern regarding the sample's representativeness, particularly the discrepancy in the number of boys and girls and their ages. It is worth noting that there is a well-known gender gap in autism spectrum disorder (ASD), with a higher prevalence among males, as well as a greater number of diagnoses and stability of diagnosis in older children, that might have contributed to this discrepancy. These factors are consistent with existing literature and may have influenced our sample composition. However, we will include a justification for these discrepancies in our manuscript to address this concern adequately, which clearly is a limitation for the study. We have included a dedicated “Limitation” paragraph to discuss these points more clearly.

My main concerns regarding the manuscript are:

- No limitations or future lines of research are included in the discussion.

  • We have included an appropriate paragraph in the Discussion section.

- The tests' reliabilities are not reported in the Measures section.

  • Thank you for mentioning this issue. We have included information on the reliability of the measurement tools used in the study in the Methods section after the description of each test.

Other minor concerns are:

- In the keywords, I would replace physical health with somatic complaints, and preschool, toddler, and children with preschoolers or preschool children.

- In Measures, the reference Schalock and Verdugo (2002) does not comply with the citation style and does not indicate the 4-point range of scores (1-4?, 0-3?).

- Regarding the variables, I recommend homogenizing the use of gender or sex, but not using them interchangeably.

- In Results, I miss a table with the results of the MLR that allows us to assess the weight of the predictor variables.

- In lines 241-242 it would be good to comment on the most significant correlations.

  • We have made the suggested changes to the keywords, regarding the references, and score range; we have homogenized the use of gender or sex terminology throughout the manuscript. In response to your suggestion, we have also included a table with the results of the MLR to facilitate the assessment of predictor variables' weights. Moreover, we have highlighted the most significant correlations to provide additional clarity.

Thank you for the suggestions and the feedback.

Round 2

Reviewer 3 Report

Comments and Suggestions for Authors

The authors have adequately addressed all of my comments.